# Soluble CD26: From Suggested Biomarker for Cancer Diagnosis to Plausible Marker for Dynamic Monitoring of Immunotherapy

**DOI:** 10.3390/cancers16132427

**Published:** 2024-06-30

**Authors:** Martin Kotrulev, Iria Gomez-Touriño, Oscar J. Cordero

**Affiliations:** 1Centre for Research in Molecular Medicine and Chronic Diseases (CiMUS), University of Santiago de Compostela, 15782 Santiago de Compostela, Spain; m.kotrulev@usc.es (M.K.); iria.gomez.tourino@usc.es (I.G.-T.); 2Health Research Institute of Santiago de Compostela (IDIS), 15706 Santiago de Compostela, Spain; 3Department of Biochemistry and Molecular Biology, University of Santiago de Compostela, 15782 Santiago de Compostela, Spain

**Keywords:** biomarker, monitoring, DPP4 (dipeptidyl peptidase 4), CD26, sCD26 (soluble CD26), immunotherapy, ADA (adenosine deaminase), chemokines, T cells, tumor cells

## Abstract

**Simple Summary:**

Prognostic markers have become a promising tool for predicting response to a certain treatment and qualifying patients for a certain therapy; however, it is important to consider that they generally describe a probabilistic scenario. For instance, the analysis of programmed death ligand 1 expression on tumor cells or tumor mutational burden is commonly employed as a biomarker with relative predictive value for immunotherapy efficacy in non-small-cell lung cancer (NSCLC) patients. Cutting-edge techniques, such as next-generation sequencing of tumor tissue and flow cytometry analysis of lymphocyte subpopulations in the peripheral blood of patients, are used with the aim of increasing the predictive value, avoiding statistical weakness and discovering algorithms based on many biomarkers. To resolve the issues posed by these novel techniques, dynamic monitoring of biomarker changes can indicate resistance or response to immunotherapy during different points of the treatment. Hence, clinicians can flexibly modify the therapy course for the benefit of the patient, improving response rates, efficiency in laboratory routines, and the economic cost of the technique. Immune checkpoint immunotherapy alters the local (solid tumor infiltration) and systemic balance among several immune system cell populations. The usefulness of immune-cell-related soluble CD26 and its DPP4 activity in immunotherapy monitoring is discussed.

**Abstract:**

Soluble CD26 (sCD26), a glycoprotein with dipeptidyl peptidase (DPP4) enzymatic activity, can contribute to early diagnosis of colorectal cancer and advanced adenomas and has been studied, including for prognostic purposes, across various other types of cancer and disease. The latest research in this field has confirmed that most, though not all, serum/plasma sCD26 is related to inflammation. The shedding and/or secretion of sCD26 from different immune cells are being investigated, and blood DPP4 activity levels do not correlate very strongly with protein titers. Some of the main substrates of this enzyme are key chemokines involved in immune cell migration, and both soluble and cell-surface CD26 can bind adenosine deaminase (ADA), an enzyme involved in the metabolism of immunosuppressor extracellular adenosine. Of note, there are T cells enriched in CD26 expression and, in mice tumor models, tumor infiltrating lymphocytes exhibited heightened percentages of CD26+ correlating with tumor regression. We employed sCD26 as a biomarker in the follow-up after curative resection of colorectal cancer for the early detection of tumor recurrence. Changes after treatment with different biological disease-modifying antirheumatic drugs, including Ig-CTLA4, were also observed in rheumatoid arthritis. Serum soluble CD26/DPP4 titer variation has recently been proposed as a potential prognostic biomarker after a phase I trial in cancer immunotherapy with a humanized anti-CD26 antibody. We propose that dynamic monitoring of sCD26/DPP4 changes, in addition to well-known inflammatory biomarkers such as CRP already in use as informative for immune checkpoint immunotherapy, may indicate resistance or response during the successive steps of the treatment. As tumor cells expressing CD26 can also produce sCD26, the possibility of sorting immune- from non-immune-system-originated sCD26 is discussed.

## 1. Introduction

Age-based screening programs, including endoscopy and other image techniques, are used to reduce mortality from colorectal cancer (CRC), breast cancer, and other solid cancers. However, this approach shows issues that could be partially solved with the analysis of biomarkers. For instance, fecal hemoglobin immunodetection (FIT) is a first line biomarker for CRC and polyp screening before colonoscopy, providing economic benefits. It is also widely known that blood biomarkers from liquid biopsy and other minimally invasive techniques offer additional information. Even though the physiological role of serum/plasma soluble CD26 (sCD26) protein with dipeptidyl peptidase 4 (DPP4, EC 3.4.14.5) activity as a ligand remains unclear (several possibilities have been reported in recent years), increasing evidence suggest that it can work as a biomarker in cancer.

## 2. Basics of the Proteins CD26 and Soluble CD26 (sCD26)

DPP4 activity on the surface of enterocytes was described in 1966 by Hopsu-Havu and Glenner [1] when studying the process of dietary protein digestion. Subsequently, in 1977, Schrader and Stacy discovered the adenosine deaminase (ADA) binding or complexing protein (ADAbp, ADCP) [2], corresponding to the DPP4 gene product transmembrane glycoprotein of 110 kDa MW, with 766 amino acids, which is usually found in a dimeric form (220 kDa). ADCP is expressed constitutively on many cell types, such as prostate, kidney, liver, and other epithelial cells, predominantly in exocrine glands and absorptive epithelia, but also on some endothelial cells of circulatory system vessels and capillaries [3,4,5,6,7]. This exoprotease belongs to the subgroup of prolyl oligopeptidases with N-terminal X-Pro enzymatic cleaving activity and regulates chemotactic responses through the cleavage of many biologically active peptides such as inflammatory chemokines CCL, 3–5, 11 and 22, and CXCL, 2 and 9–12 [7,8,9,10]. In addition, it can split incretins of the insulin-related metabolism and neuropeptides such as NPY or VIP [11,12]. Further to its enzymatic activity, ADCP is also a functional receptor for extracellular matrix (ECM) collagen and fibronectin for plasminogen and caveolin-1 [7,11,12].

In 1984, Fox et al. [13] described the protein as a leucocyte antigen which binds to the Ta1 monoclonal antibody. In 1993, this protein was ascribed to the CD26 cluster independently by the groups of Houghton and Schlossman [14,15]. CD26 expression in leucocytes is not constitutive but is low in the resting state of T and NK cells and rapidly upregulated upon activation [4,12,16,17].

Serum DPP4 activity was first discovered in 1968 by Nagatsu’s group in Japan [18]. Within human plasma/serum, 90–95% of DPP4 activity has been associated with the concentration of soluble (in contrast to transmembrane) CD26 (570 μg/L), known as sCD26 [5,6,7,11,19,20,21,22]. sCD26 lacks transmembrane and cytoplasmic domains as its amino acid sequence starts at the 39th position of membrane CD26; nevertheless, in its glycosylated state, its molecular weight is analogous to transmembrane CD26 [21,23]. sCD26 can also be found in most other biological fluids [5,7,19].

Apart from its DPP4 activity, there have been interesting advances on the physiological role of sCD26 as a ligand. sCD26 acts as a chemorepellent of neutrophils, cells that are implicated in cancer, through its binding to PAR-2 (protease-activated receptor-2), a G protein-coupled receptor present in many different tissues [24]. In the rat model of HCC, where a high-fat diet enhanced the sCD26 levels, this protein promotes inflammation and insulin resistance through PAR-2 in cooperation with plasma factor Xa [25]. The first activates the caveolin-1 (CAV1)–interleukin-1 receptor kinase (IRAK1)–TGF-β activated kinase 1 (TAK1) pathway, while factor Xa activates the PAR2-RAF1 pathway. Both acting in synergy induce monocyte chemokine protein 1 (MCP1) and IL-6 secretion via adipose tissue macrophages [26]. 

sCD26/DPP4, as well as other chymotryptic serine proteases such as prostate-specific antigen (PSA), high-temperature requirement protein-A (HtrA), or the bacterial virulence factor subtilisin, can activate human monocyte-derived macrophages leading the expression of pro-inflammatory cytokine genes IL1B and IL6 and the indoleamine-2,3-dioxygenase (IDO1) protein [27]. IDO1 and the cytokines trigger the activation of the NF-kB pathway, which is also activated by the PAR-2 pathway [28]. In addition, IDO1 enzymatic activity enhances the kynurenine pathway, considered a major checkpoint in cancer development, by modulating tolerogenesis in the immune system.

We previously described that patients with angiodysplasia exhibit higher sCD26 and soluble DPP4 activity titers [29]. The analysis of endothelial cells isolated from patients demonstrated an enrichment of CD26 and PAR-2 [19], which are involved in enhanced angiogenesis [30,31]. Furthermore, sCD26 induced CCL2 expression [26]. However, the DPP4 activity can cut proangiogenic chemokines [32,33,34], and this unresolved issue may be important in the tumor development [10].

PM CD26 (probably together with ADA) is a plasminogen receptor [35] which is soluble, constituting large complexes with AntiCD26 auto-antibody and bound with CD26 (unpublished data). Therefore, this context of coagulation/fibrinolysis deserves further attention.

## 3. CD26 in Solid Tumors and Why sCD26 Was Studied as a Cancer Diagnostic Biomarker

When CD26/DPP4 was known as ADCP [3,36,37], low levels of solubilized CD26 were found in total cancer-derived cell lines and homogenates of colon, kidney, lung and liver tumors [3,14]. Since then, it has consistently been associated with cancer, and its involvement in tumorigenesis has been extensively studied [38]. Depending on the tumor microenvironment, CD26 can act both as a tumor suppressor or activator. Since the protein is involved in cell-to-cell adhesion through ADA, altered expression of CD26/DPP4 is capable of inhibiting beta integrin-1 and ECM synthesis. Moreover, DPP4 was found to be engaged with other CD26 homologous cell surface proteases, such as MMPs and FAPα, and expressed in tumor cells, malignant transformation, and cancer progression, thereby facilitating invasion [39,40,41,42]. Digestion of ECM components may promote the passage of malignant cells through basement membranes and stromal barriers.

Despite the first study on colorectal cancer (CRC) reporting reduced levels of enzymatic activity in a small group of patients [43], other studies found increased DPP4 activity [44]. We performed a comparative analysis of sCD26 secretion in healthy donors and CRC patients via ELISA [45]. The subsequent outcomes showed a reduced sCD26 levels in patients [45], lower in the early stages of the disease. Consequently, the sensitivities of sCD26 were higher than 80% in Dukes’ stages A, B and C (Figure 1A), whereas in Dukes’ stage D, carcinoembryonic antigen (CEA) levels exhibit a higher sensitivity than sCD24. This fitted with the study mentioned [44], as it was in stage D that DPP4 activity levels increased. In this study, sCD26 as a variable was not related with tumor variables such as Dukes’ stage classification, age, sex, tumor location or degree of differentiation, suggesting the potential use of sCD26 for CRC early diagnosis. With a follow-up of 2 years until recurrence, the data also showed a preliminary potential prognostic value. Regarding the specificity, no changes were found in gastric tract carcinomas. In some blood cell cancers, the concentration levels were raised, but this variation can be potentially influenced by benign pathology of the gastric tract and Crohn’s disease [45,46].

Focusing on its influence on metastasis, the outcomes fit with current evidence that cancer stem cell (CSC) subsets expressing CD26 are implicated in metastases of many cancers [38,47,48], including hematological [49], and the expression level of CD26 in these cells is associated with the prognosis of the disease [49]. The CD26-expressing cells of malignant pleural mesothelioma (MPM) upregulate the secretion of periostin, a protein that enhances the migration and invasion of MPM cells [50]. DPP4 has been related to the epithelial–mesenchymal transitions [47] and the chromatin remodeling axis that leads to liver metastasis [51]. Our recent results demonstrated that CD26 is involved in cell-to-cell homotypic aggregation in spheroids [48]. This fact might be related to the collective cell, rather than single cells; more efficient migration of highly metastatic cells that form multicellular homo- and heterotypic aggregates; and resistance to anoikis and adhesion to the endothelium (and other cells) within the metastatic site.

Of note for subsequent discussion, it was recently shown that DPP4 is also redistributed intracellularly towards the nucleus in a tumor-suppressor p53-dependent way [52]. In hepatocellular carcinoma (HCC), the subcellular redistribution is lost due to the lack of p53, leading to lipid peroxidation via plasma membrane (PM) DPP4 interactions with NADPH oxidase 1 (NOX1) (an interaction that can be blocked by DPP4i [26]), ultimately resulting in ferroptosis, a protective process in cancer patients. DPP4 has also been related to the Wnt axis of ECM remodeling [53] and other intracellular targets related to cell regulation [54]. Wnt recognizes a heparan sulfate structure on GPC3, where CD26 can be bound, losing its DPP4 activity (in vitro [55]). One proposed mechanism suggests that glypicans behave as co-receptors which bind both the Wnt receptor and its ligands. 

To validate sCD26 concentration as an early biomarker for CRC, using several studies, we found that sCD26 was informative of advanced adenomas, colorectal polyps in general, grade of dysplasia, diverticula and hemorrhoids, and of smoking status [29,56]. Patients with other tumors have been studied; the serum DPP4 activity and overall percentages of CD26+ lymphocytes and CD26+ white blood cells have been measured in patients with hematological malignancies such as non-Hodgkin lymphoma (NHL), Hodgkin lymphoma (HL), leukemia, plasmacytoma, and multiple myeloma. In patients with NHL, leukemia, multiple myeloma, and melanoma (more so than vitiligo), significantly decreased DPP4 activity and percentages of CD26+ lymphocytes and overall CD26+ white blood cells and lymphocytes have been observed in comparison with healthy individuals. Many malignant breast cancer patients in early stages exhibit higher DPP4 activity than controls [57,58,59,60].

In fact, the accumulated information from the past 25 years supports that low levels of DPP4/sCD26 (in some hematological and solid malignancies) occur concurrently with impaired immune status, whereas increased levels occur with enhanced immune status in inflammatory and infectious diseases, other hematological tumors, and liver diseases [11]. We also cite many putative explanations of the lack of a strong correlation between this DPP4 activity and the sCD26 protein concentration [7,11], and both have been used together or separately to show the potential utility of this protein as a marker in the screening, monitoring, and prognosis of some cancers [11,19].

It is also worth mentioning that in our recent article [29] studying a cohort of 1703 individuals who underwent a colonoscopy and had a serum sample, sex and age differences for both activity and protein titers were found. As in that study, individuals with colorectal cancer and advanced adenomas were included; in Table 1, those cases are excluded, and it emerges that the resulting cohort includes no-neoplasia benign findings (as would have a standard healthy population of a certain age).

## 4. Expression of Cell Membrane CD26 on Leukocytes

Originally, CD26 was considered an activation marker [61] since its expression on lymphocytes is closely associated with different CD45R isoforms that define the effector/memory phenotype of helper T cells [15,16,17,62]. Those analyses have improved over time with the discovery of different anti-CD26 antibodies.

First, it was observed that all CD4+ CD8+ medullary thymocytes express CD26 [63]. Moreover, 90% of human cord blood T cells (almost entirely CD45RA+, naïve) are CD26+ [61]. Later, it was found that the frequency of CD26+ T cells is much lower in adult blood and within lymphoid tissue [64,65]. All of this suggests that CD26 expression can also be suppressed as T cells differentiate, i.e., with the presence of subsets of CD4 or CD8 CD45R0 CD26-negative T cells [66,67,68,69,70]. This is the case of Tregs [71] and CD4 and CD8 central memory T cells [65], populations with clinical implications.

B cells, natural killer cells (NKs), and monocytes are CD26-negative under basal conditions, but CD26 can be upregulated upon activation [72]. Regarding dendritic cells (DCs), pDCs demonstrate the highest levels of CD26 expression [73]. Moreover, MAIT and NKT cells also express high levels of CD26 [74,75]. CD26-rich NKT cells upregulated TIM-3 expression in COVID-19 patients, inducing their depletion and dysfunction. T cell immunoglobulin and mucin domain 3 (TIM-3) form an inhibitory immunocheckpoint (ICP) that belongs to the TIM gene family, so a future ICP therapy involving TIM-3 is expected.

The recent analyses of the CD4 CD26-high T cell subset showed that it is composed of Th1, Th17 and hybrid Th1/Th17 cells, all capable of transendothelial migration [76,77,78,79] and related to clinical severity in multiple sclerosis [80] and rheumatoid arthritis [67,68]. However, we recently described that there is also a Th2 CD26-high subset and Th2 with lower expression [65]. Cell phenotype may have clinical consequences in immunotherapy, because it has been described that human CD4 CD26-high cells have an enhanced chemokine receptor profile and stemness, are more resistant to apoptosis, more cytotoxic and persistent when they are adoptively transferred, and are capable of causing multiple solid tumors to regress [76]. For that purpose, CD26-high T cells have been engineered with a mesothelin-specific chimeric antigen receptor (CAR), which ablate large human tumors more than CAR-T subsets enriched in Th17, Th1, or Th2 cells [77].

The same group has been recently analyzing some robust and reproducible biomarkers that could be validated to select patients with metastatic melanoma for treatment with ICI nivolumab (an anti-PD1 antibody), as a large percentage of them show resistance to PD-1 inhibition and occasionally experience severe off-target immune toxicity [81]. The frequency of circulating CD4+CD26-high T lymphocytes is a potential biomarker since it is associated with clinical outcomes (and at least with survival in melanoma patients). All melanoma patients had reduced CD4+CD26-high T cell frequencies compared to healthy subjects, while low baseline percentages of CD4+CD26-high T cells were associated with worse clinical outcomes. The treatment enhanced these percentages, and patients with clinical benefit from nivolumab therapy had significantly higher frequencies of circulating CD4+CD26-high T cells than patients with non-clinical benefit [81].

Other studies [82,83,84,85] in melanoma patients have reported that a higher frequency of circulating central memory T cells (CD4 and CD8) that are CD26-negative [86] is associated with an increased tumor inflammatory profile, with longer survival times, and with clinical response to anti-CTLA-4 treatment.

In addition to its DPP4-cleaving activity, it was proposed that CD26 participates in the regulation of immune function after some studies observed that certain anti-CD26 monoclonal antibodies (mAbs) and ADA as a ligand, act as co-activating signals of T cells [19]. CD26 can also play a key role in the immunological synapse through ADA binding to the adenosine 2-Receptors and/or to caveolin-1 in APCs. It is possible that integrin beta-1, the T cell transmembrane tyrosine phosphatase CD45, Mannose 6-Phosphate/Insulin-like Growth Factor II receptor (M6P/IGF IIR), the chemokine receptor CXCR4 and ECM glypican-3 (GPC3), and even plasminogen [11,12,19,64,65,66,69,77,87] are also involved. As mentioned, through its binding to collagen and fibronectin in the extracellular matrix, CD26 could participate in T cell infiltration, as could CD26+ metastatic or blood-born cancer cells [88]. Integrin beta-1 and the MAPK pathway were implicated in the mechanism. A similar pathway was formulated for lymphocyte interaction with the endothelium, although in this case, LFA-1, an integrin beta-2, is implicated [86]. However, the consequences of higher or lower CD26 expression in the different T cell subsets remain to be elucidated. Some pathogenic molecules, including virus spike proteins, can bind to CD26 as well [88]. Our observation of CD26 involved in the aggregation of CD26+ tumor cell lines, which can have a significant role in metastasis [48], has not been tested in immune cells. 

Although CD26 is a marker of senescent cells [12,89], some CD26-high cells are exhausted [76], and most of them can persist in solid tumors [77]. It has been recently demonstrated in fact, in both the CD4 and CD8 compartments, that antigen-experienced, polyclonal and exhausted cells constitute a fraction of CD26-negative cells [90,91,92]. Importantly, these cells are a main contributor to the very high response rates to immune checkpoint inhibitors in hematological tumors and probably also in solid tumors, as the exhausted cell state is characterized by elevated co-expression of PD-1, CTLA-4, TIM-3, and LAG.

## 5. sCD26 from Leukocytes in Relation to Specific Populations

The kidney and the hepatobiliary system were the first proposed obvious potential sources of sCD26, harboring large amounts of PM CD26. However, anephric individuals have normal amounts of sCD26 with approximately two-fold more sialic acid than kidney CD26 [93,94,95,96,97,98,99,100], so the kidney was discarded as the main source [21]. Regarding the hepatobiliary system, even though the brush border of hepatocytes may secrete sCD26 under specific conditions [98], CD26 is predominantly located in the bile canaliculi [98,101,102]. Moreover, DPP4 activity levels did not correlate with other markers of bile duct or hepatocyte injury in chronic hepatitis C and other liver viral infections [103]. The endothelium of venules and the capillary bed of several organs, the lungs, myocardium and striated muscles, spleen or pancreas can also shed sCD26 [4,7,46,95].

In some of those studies, some pathologies showed increased serum DPP4 activity [98,102,103], and it was suggested that the new protein could be shed from the peripheral blood T cells directly from those involved in the control of the viral infection or indirectly by stimulating hepatic stellate cells. In fact, T cells had already been suggested to be involved in liver regeneration [99], and Kasahara et al. [104] had identified them in serum isoforms not only from the liver, spleen, or kidney but also from the immune system.

The data acquired from numerous studies provide clear evidence that most serum CD26 production is primarily attributed to T cells [11,46,95]; however, the relationship between the different isoforms of the protein and diseases remains unclear [104,105], and the fraction of immune system-originated sCD26 can be regulated [23,106,107]. With the commercialization of sCD26 ELISAs at the turn of the millennium and the knowledge gained from the recorded data of CD26 expression on T cells, sCD26 was used in many studies as marker of Th1 cellular immune activation, in many with sCD30 or sCD23 as markers for Th2 (humoral response). The sCD26 titers increased in HIV-1, leishmaniasis, myocardial infarction, and atopic dermatitis patients; in asthmatics, osteoarthritis and gastric cancers did not change; in rheumatoid diseases, mostly in lupus erythematosus and Sjögren syndrome, the level decreased, whereas the results from hepatitis C virus (HCV) were inconsistent [108,109,110,111]. Moreover, the loss of CD26 expression in the disease progression of cutaneous T cell lymphoma (CTCL) and its most frequent forms, mycosis fungoides (MF) and Sézary syndrome (SS), correlated with lower serum CD26/DPP4 levels. This lower activity induced the migration of SS cells due to a reinforced chemoattraction [49,107], which it will be explained later.

Even so, it was not until 2018 that it was demonstrated, both in humans and in mouse models, that lymphocytes are a major source of circulating soluble DPP4. This was evidenced in individuals with congenital lymphocyte immunodeficiency following successful restoration of lymphocyte hematopoiesis, as well as in irradiated lymphopenic mice and wild-type to DPP4−/− animals. The study showed that activation of T lymphocytes resulted in increased levels of sDPP4 and that acute viral infection induced a transient rise in sDPP4, which correlated with the expansion of antigen-specific CD8+ T cells [112]. 

In fact, the levels of DPP4 enzymatic activity and/or sCD26 serum levels correlated with specific T cell subsets [68,70,111]. In multiple sclerosis, after in vitro stimulation of PBMCs, cell proliferation correlated with serum sCD26 levels [70]. Serum DPP4 activity, more than sCD26 expression, was found to be impaired in rheumatoid arthritis patients compared to healthy controls. Levels of sCD26 correlated with the number of CD4 CD45R0+ CD26-negative cells, with the population showing the most significant increase in these patients, which mainly comprises TCM and Treg subsets. Additionally, sCD26 levels correlated negatively with cell surface CD26 MFI on the CD4 CD45R0-CD26+ population (CD4 T naïve cells) [68]. With the aim of extending this analysis, our most recent assays identified a positive association between sCD26 and CD8 CD45R0+ CD26-negative subset, plasmacytoid DCs, the count of CD8 CD45R0+ CD26-high % (polarized T cells), CD26 MFI of CD8 CD45R0− CD26+ and different subsets of CD26+ monocytes’ frequencies. However, a negative relation was observed with the frequencies of naïve CD8 CD26+, CD8 CD26-high and the CD4 T cell CD45R0+ CD26-high subset (not published).

These results suggest that there may be different subsets and different mechanisms of sCD26 production implicated, but changes in circulating leukocyte subsets are reflected by changes in sCD26 levels and DPP4 activity.

There are several studies showing in vitro results that substantiate this proposition. After polarization to Th1, Th2 and Th17, a significant amount of sCD26 in the secretomes of CD4 T helper subsets was noted, but the levels of sCD26 were lower compared to the Th0 condition (activation without polarization) [63]. In accordance with very early data on soluble CD26 found in the lumen of secretory granules of endocrine cells [113,114,115], CD26/DPP4 is also stored in secretory granules of several major human cytotoxic lymphocyte populations, NK and CD3+/TCR γδ+ cells, and also some αβ+ CD4 and CD8 cells, specifically CD8-low. Upon stimulation, in a Ca2+-dependent manner, degranulation leads to a massive release of proteolytically active sCD26/DPP4 together with other effector proteins such as granzymes, perforin, and granulysin [116].

## 6. Studies That Support Changes in sCD26 Levels as a Tool for Patient Monitoring

For a CRC follow-up study [117], sCD26 together with clinical biomarkers such as CEA, carbohydrate antigen values of 19.9 and 72.4 were measured in 43 patients with primary CRC. The pre- and post-operative serum samples were obtained for surveillance over an average follow-up period of 41.8 ± 20.8 months. Patients were grouped as without disease (*n* = 28), with tumor persistence (*n* = 2), with local recurrence (*n* = 3) or distant metastasis (*n* = 10). We observed that sCD26 levels showed well-defined patterns during follow-up, where disease-free patients had stable levels between 460–850 ng/mL, while high (over 850 ng/mL) and unstable sCD26 levels were found before recurrence was diagnosed (Figure 2). 

When we calculated the ratios of maximum to minimum sCD26 values during surveillance, the means were 1.52, 2.12, and 2.63 for patients with no recurrence, local recurrence, and metastasis, respectively (Figure 3).

In 2014, a study showed that only the group of patients with multiple sclerosis that responded long-term to IFNβ therapy showed similar DPP4 activity to the group of healthy controls [70]. Previously, it had been reported that response to interferon plus ribavirin therapy in patients with chronic hepatitis C correlated with changes in soluble CD26 and CD30 levels [109].

In a similar cross-sectional case–control study with rheumatoid arthritis patients [68], we examined the levels of serum sCD26 and DPP4 activity in patients’ groups defined according to the therapies (conventional or biological) with anti-TNF, anti-CD20, anti-IL6R, or Ig-CTLA4. Information on the disease score and the inflammatory status of the patients involved in the study is shown in Table 2 (published in [68]).

We observed relevant differences; the anti-TNF group showed higher DPP4 activity than the group under classical DMARDs, and the anti-CD20 group showed lower sCD26 levels. The cases with Ig-CTLA4 were a few (four), but it was remarkable that whereas the sCD26 titers were low but comparable with those of cDMARDs, the DPP4 activity levels were very low (about half of those of the anti-TNF group) (Figure 4).

In two cohorts of Crohn’s disease [CD] and ulcerative colitis [UC] patients recruited in a prospective study, a sub-cohort was evaluated using clinical indexes and followed up to assess for treatment escalation [118]. Patients were enrolled in the induction and maintenance phases of biological treatment (different anti-TNFs, or vedolizumab, an anti-integrin α4β7 Ab that inhibits T cell binding to adhered Ag MAdCAM-1 expressed on gut cells) and evaluated at several time points (five per patient). Median DPP4 levels were significantly lower in active IBD patients when compared with those in remission (lower levels in CD than in UC). In fact, DPP4 was able to distinguish clinical and endoscopic activity from remission and also from the need for treatment escalation. In the follow-up, DPP4 levels were higher in responders to treatment and more pronounced among UC. In the study, fecal calprotectin (FC) and CRP were also analyzed, and the three biomarkers had a similar ability to distinguish between IBD responders and non-responders. CRP and ferritin can identify which patients before treatment are at high risk of experiencing side effects and poor outcomes, such as reduced progression-free and overall survival after CD19-targeted CAR T-cell therapy for hard-to-treat B-cell lymphoma [119]. These results altogether suggest that DPP4/sCD26 titers reflect the response to therapy, although in part, titers may also account for pre-treatment individual responsiveness. In the one follow-up study of cancer under biological therapy that we know of, a phase I trial with humanized anti-CD26 mAb [120], serum sCD26/DPP4 titers were reduced following YS110 administration in patients with solid-tumor renal cell carcinoma (RCC), malignant mesothelioma (MM), and one urothelial carcinoma (UTC) and later gradually recovered until the next infusion. Both effects, i.e., the dropping and the raising of titers, were clearly visible in three infusions in a month. Moreover, the titer before the next infusion was sustained at lower levels in stable disease than in progressive disease cases. This state was defined by the tumor volume variation using the RECIST (response evaluation criteria in solid tumors) criteria two weeks after the last mAb infusion. In this work, no data on CD26+ immune cells were shown.

For comparison, in a follow-up study of hepatitis C virus patients undergoing antiviral treatment with pegylated interferon and ribavirin, no difference in DPP4 activity was found in patients with therapeutic non-response [121].

Some additional data with mouse models undergoing biological therapies can be found in a recent review focusing on DPP4 in autoimmunity [72].

## 7. sCD26 Shedding and/or Secretion: Biochemistry of sCD26

The finding of intracellular sCD26 stored in granules of some T cell subsets was not a surprise per se because very early on, CD26-negative T cells showed intracellular CD26 staining [122], and CD26+ exosomes had also been described [82]. In fact, there is a relationship between many classes of effector vesicles, most of them containing CD26/DPP4 [123]. 

Similarly, because there are so many types of cells expressing cell surface CD26, numerous proteases mediating the release of sCD26/DPP4 have been reported—matrix metalloproteases 1 (MMP1), MMP2, MMP9, MMP10/13, and MMP14, depending on the cell tissue [8,11,13], or the serine protease kallikrein 5 (KLK5) in Th17 cells [19,124], which are likely differentially regulated [125]. Proteolytic cleavage by KLK5 active sites resides at region 38–40 of DPP4.

Two CD26 forms were detected in mice [126], and we found two mRNAs in humans [127], but we do not know if these or the genetic variants of DPP4 (which were linked to clinicopathologic development of prostate cancer [128]) are involved in these differences.

Interestingly, in vitro and in vivo studies identified HIF-1α-dependent strong upregulation of DPP4 mRNA expression under hypoxia growth in cancer cell types including smooth muscle cells, adipocytes and ovarian cells, and the increased protein levels led to its proteolytic shedding from the cell surface in an inactivated form mediated by at least MMP10 and MMP13 [129,130,131]. It has been also reported in hepatoma cells that Serpin B3 induces overexpression of inactive DPP4 [132]. These findings suggest an additional reason that activity is often unrelated to serum titers.

However, we want to highlight here the possibility that many different forms of circulating sCD26 exist in blood, each of them related with different types of cells and that eventually could be sorted into more specific biomarkers. For example, the proteolytic cut by KLK5 fits well with the original characterization of serum sCD36 starting at the 39th residue from the total 766 amino acids of each protomer [22,124]. However, the semen sCD26 is isolated with the same mAb start at the 30th residue [48]. Additionally, the protein glycosylation analyzed in many tissues is different, the serum form being highly sialylated (a type of glycosylation) [23,82,83,95]. Sialylation is strongly enhanced in elderly individuals, which can affect DPP4 activity [133,134], consistent with the fact that serum/plasma DPP4 enzymatic activity decreases with age [5,29]. However, there are neither biochemical analyses of the secreted sCD26 nor glycomic studies of the leukocytes’ PM CD26.

When we analyzed serum sCD26 by Western blotting, different bands could be appreciated with different anti-CD26 Abs (see Figure 5 as an example). In many secretomes or cell lysates analyzed from many solid tumor lines, bands with different molecular weights were seen [135]. It can be expected that the percentage of different ligands described for CD26 can change with disease. Advances in this knowledge can have impact in the design of more specific ELISAs for the study of the biomarker.

## 8. CD26 Substrates and Inhibitors

Whereas other proteases involved in cancer development and progression are mainly degrading ECM proteins, the DPP4 enzymatic activity of CD26 shows a different character, modulating the biological activity of many regulatory peptides and, at least, participating in systemic homeostasis. Consequently, DPP4 has become a relatively novel therapeutic target for many inhibitors (DPP4i), which prolong the half-life of endogenously produced insulin in diabetics [10,11,12,16,19,72] by preventing the degradation of incretins secreted by the enteroendocrine system [136]. It is important to take into consideration that DPP4i can affect the collagen-matrix-dependent activity related to the motility capacity of both immune and tumor cells. For patients on immune checkpoint inhibitors, it must be checked whether they are undergoing therapy using these inhibitors, with the understanding that we want to measure DPP4 activity levels in addition to sCD26 titers, remembering that additional information can be obtained by studying their correlation [29].

It is worth noting that the half-life of other growth factors and chemokines is also regulated by sCD26/DPP4 [7,10]. Given that immune checkpoint inhibitors and VEGF inhibitors, already a first-line combination for systemic treatment of some cancers including HCC, influence the infiltration of T cells into tumors, and several chemokines are involved in this trafficking, DPP4i could serve as potential adjuvant to therapeutic agents. Some groups have already developed preclinical studies with those same inhibitors [26] for HCC and ovarian cancer [137,138]. DPP4i enhanced antitumor effects by increasing the T cell trafficking through the preservation of biologically active CXCL10. CXCL12, the ligand of CXCR4, and the other CXCR3 ligands (CXCL 9–11) might also be affected as they are converted into antagonists upon DPP4-dependent truncation in vitro. The inhibitors also enhance the role of CCL11-dependent eosinophils on antitumor immunology. However, by preventing the cleavage of MIP-1alpha, CCL3, they deregulate M1/M2 macrophage polarization, attenuating anticancer immunity. To note, DPP4i may have other activities via caspase-1 activation or inhibiting metastatic traits through opposite effects to those of TGF-beta1 [48] and bind to CD26+ leucocytes with unknown consequences.

For the purpose of this Special Issue, we note here that some of the chemokines truncated by DPP4 can be measured. CXCL10 can be processed to an inactive form in ovarian cancer, and the group of Stephens et al. evaluated the ratio of active to total CXCL10 to improve the identification of malignancy. Recently, the addition of this test to a biomarker panel—also including pro-inflammatory IL6, cancer antigen 125 (CA125) and Human Epididymal Protein 4 (HE4) measurements, which are part of the Risk of Malignancy Index (RMI) or Risk of Malignancy Algorithm (ROMA) scores used to indicate likelihood of malignancy when an ovarian mass is present—strongly improved the discrimination of benign from malignant disease [139,140]. We suggest that multiplexing with this chemokine, truncated or not, and IL6 may improve the specificity of the test proposed.

## 9. ADA, a Clinical Biomarker Related to sCD26

Adenosine, a purine nucleoside that is present at increased levels in the hypoxic tumor microenvironment, has a big impact on the immune response to a cancer [141], and can downregulate CD26 on colorectal cells. Extracellular ADA, of course, can degrade it to inosine. As mentioned, CD26 was first called ADAbp or ADCP and found expressed in many tissues related to solid tumors. Additionally, we demonstrated that ADA on lymphocytes can be regulated by cytokines independently of CD26 [142]. ADA enzymatic activity is a clinical biomarker for certain diseases [143].

ADA can be found bound to sCD26 in sera [20,37]. It can be expected that titers of ADA bound to CD26 can also reflect the specificities of the sCD26’s origin, but this is unknown. It should be pointed out the site of binding to ADA in CD26 is very close to some epitopes of anti-CD26 Abs, which complicates measures in techniques that use Abs. 

## 10. sCD26 in Saliva

We have discussed elsewhere that a non-invasive method for the study of biomarkers has many advantages for screening programs [29]. Blood extraction also shows better adherence than fecal tests or imaging techniques. Obviously, saliva is a more convenient biological fluid than blood, especially in hospitals or when working with elderly people or people with mental disorders [144].

A recent study that evaluated human salivary proteases in Sjögren’s syndrome (SS), an autoimmune exocrinopathy characterized by progressive damage to the salivary and lacrimal glands associated with lymphocytic infiltration, found increased activity of human sDPP4/CD26 in SS saliva [145] and obese adults [146], although it might have specificity problems in patients with periodontitis [147,148]. As mentioned, CD26 is an exosomal membrane marker; intact exosomes could be isolated from whole saliva stored at 4 °C and kept intact after 20 months of storage at 4 °C [149].

However, to our knowledge, there is only one study of the correlation between salivary and plasma sCD26. Although in that study it was found that sCD26 levels and DPP4 activity were unaffected in postmenopausal women and women with regular cycles, saliva measurements seem to be poorer predictors than plasma ones [150].

## 11. Conclusions

The quality of the immune response at the start of tumorigenesis and its intensity in the tumor microenvironment in the late stages of the tumor progression are inversely correlated. In NSCLC patients who received nivolumab, peripheral T cell subpopulations can inform us about the state of the anti-tumor immune response [56,57,58,59].

Another way to check it is through the predictive value of inflammatory markers, and serum changes in circulating leukocyte subsets are reflected in sCD26 levels and DPP4 activity changes. Some biological therapies, including one targeting CTLA-4, alter those levels, and they have been studied as diagnostic, prognostic, and monitoring biomarkers. In some studies, there were correlations with other well-known inflammatory biomarkers such as CRP, which is already used to inform PD-1 inhibitor therapy. For example, both had a similar ability to distinguish between IBD responders and non-responders to biological therapy. sCD26 and DPP4 alongside CRP or ferritin (which are already used for risk stratification in immunotherapy) can feature in a readily available and routinary laboratory test.

Our initial hypothesis was that the impairment of circulating sCD26 titers in CRC, for example, is related to the tumor-dependent suppressive environment on lymphocytes. However, because many DPP4 reactants are chemokines and ECM proteins, this impairment in many solid cancers might be associated with the migration and infiltration of specific populations (for example, CD26-high and/or cytotoxic cells) into the tumors. It may also be associated with the stemness of these subsets and the diminution of exhaustion of the effector T cells.

The still incompletely explored binding of sCD26 to several other proteins or isoform differences might enhance the correlations of each of them to specific leukocyte subsets. Proteolytically active CD26/DPP4 is also present on plasma exosomes derived from both tumor cells and T cells, having the capacity to ligate potential interaction partners, but knowledge of them is also scarce. Finally, it would be interesting to check if saliva sCD26 could behave as biomarker for monitoring.

## Figures and Tables

**Figure 1 cancers-16-02427-f001:**
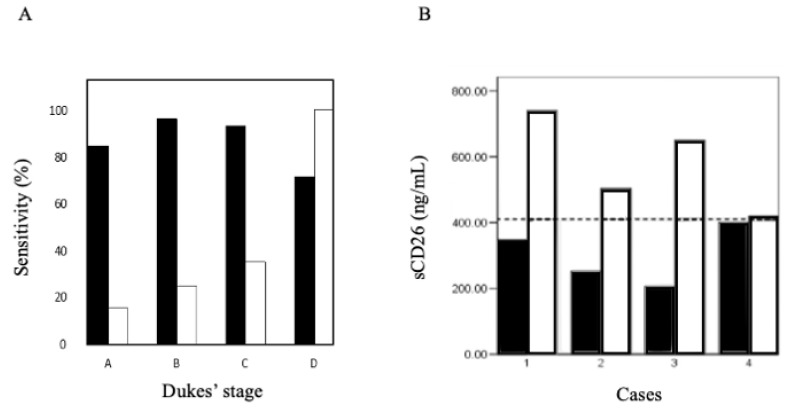
(**A**) Sensitivity of sCD26 (black boxes) and CEA serum levels for the diagnosis of colorectal cancer in the four Dukes’ stages (12, 55, 29, 14 samples, respectively) [45]. (**B**) Normalization of the sCD26 levels before (black bars) and after polypectomy (white bars) in four cases diagnosed with polyps. A 410 ng/mL cut-off point is denoted by a discontinuous line [46].

**Figure 2 cancers-16-02427-f002:**
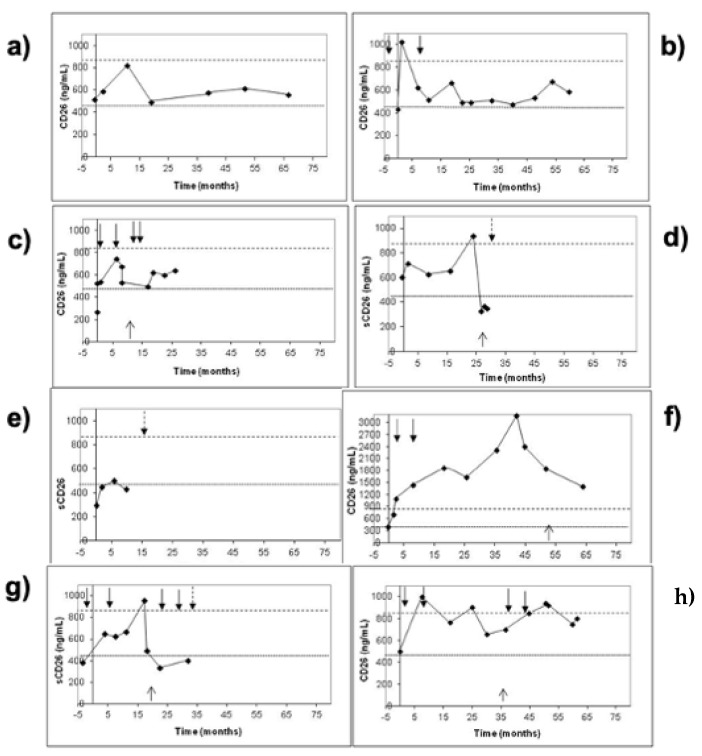
Follow-up sCD26 levels [117]. (**a**,**b**) Two representative disease-free patients (**c**,**d**) with tumor recurrence, (**e**) one representative patient with tumor persistence, and (**f**) one who developed hepatic and (**g**,**h**) pulmonary metastasis. Black arrows indicate the beginning and end of chemotherapy cycles; the upward arrow indicates diagnosis of metastasis, and the dashed arrow, exitus.

**Figure 3 cancers-16-02427-f003:**
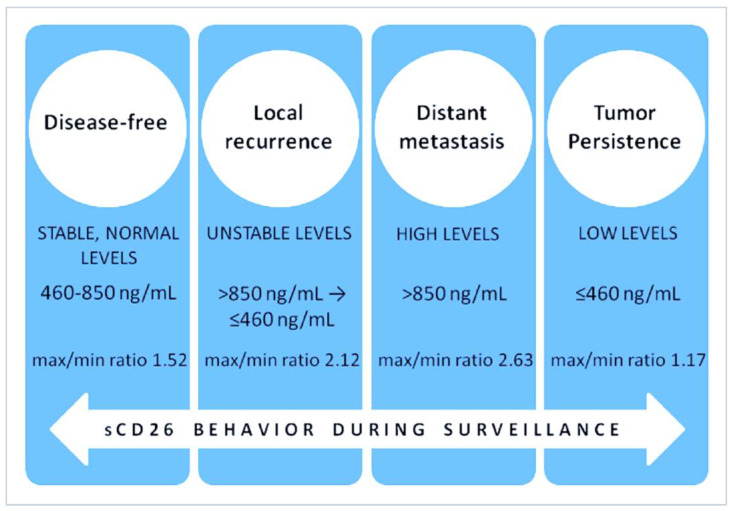
Schematic representation of the behavior of sCD26 during follow-up of CRC patients according to the disease status [117].

**Figure 4 cancers-16-02427-f004:**
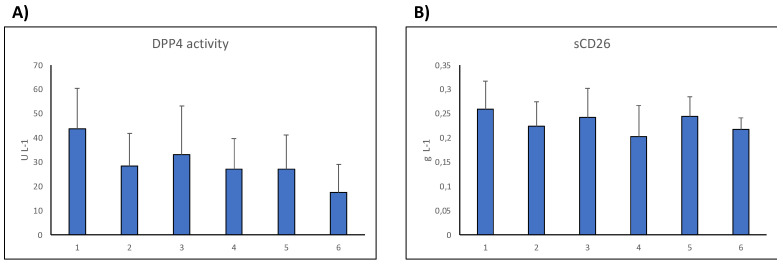
Dipeptidyl peptidase 4 (DPP-IV) enzymatic activity levels (mean and SD, U L^−1^) (**A**) and sCD26 concentration levels (mean and SD, g L^−1^) (**B**) in serum of RA patients grouped according to the type of therapy and healthy donors [88]. N = number of samples; (1) HC, healthy donors n = 25; (2) cDMARD, conventional disease modifying anti-rheumatic disease, n = 21; bDMARDs (3) anti-TNF, n = 58; (4) anti-CD20, n = 12; (5) anti-IL6R, n = 11; (6) IgCTLA4, n = 4.

**Figure 5 cancers-16-02427-f005:**
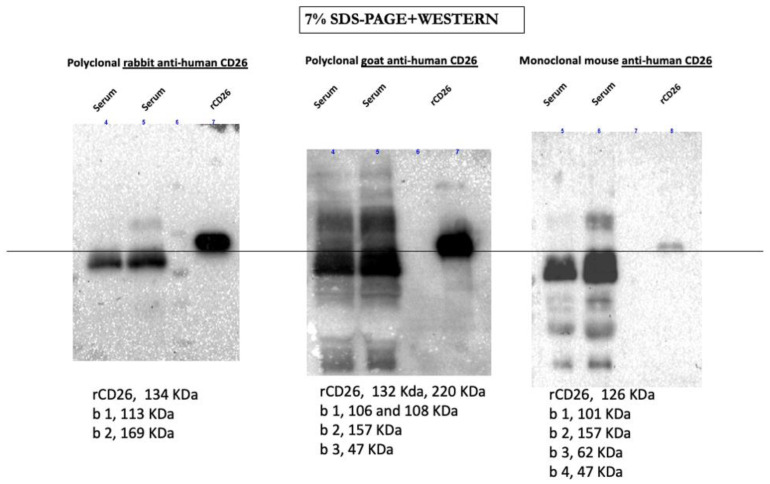
Western blot analysis of sCD26 presence in the serum of two donors. The membranes were developed with polyclonal rabbit anti-CD26 Ab ((**left**), Novus Biologicals), polyclonal goat anti-CD26 Ab ((**centre**), RnD Systems), and mAb antiCD26 ((**right**), Immunostep, Salamanca, Spain). Additionally, 20 μg of protein in each line was treated at 37 °C for 15 min before SDS-PAGE electrophoresis, (there were no recognized Ab bands from samples treated at 100 °C). Data shown are representative of at least three experiments. For comparison, the molecular weight of the main detected bands in each gel is shown below them and the black line shows the expected weight of sCD26 (Appendix A shows the original gels: Appendix A).

**Table 1 cancers-16-02427-t001:** Levels of serum DPP-IV activity and sCD26 concentration according to the demographic characteristics of the studied population.

Characteristic	DPP4			sCD26		
	N	Mean ± SD (mU/mL)	*p*-Value	N	Mean ± SD (ng/mL)	*p*-Value
Sex						
Women	372	45.98 ± 11.15		602	536.12 ± 183.09	
Men	299	40.30 ± 10.91	<0.001	470	496.27 ± 182.78	<0.001
Age (years)						
≤49	14	44.05 ± 7.92		110	523.12 ± 174.83	
50–59	285	45.48 ± 12.43		418	535.40. ± 185.73	
≥60	372	41.87 ± 10.36	<0.001	544	504.88 ± 183.56	0.006

**Table 2 cancers-16-02427-t002:** Disease activity parameters of patients grouped according to their biological (BT) or conventional (no BT) therapies.

	No BT (n = 21)		Anti-TNFα BT (n = 47)		Anti-CD20 BT (n = 10)		Anti-IL6R/Ig-CTLA4 BT (n = 13)	
	Mean ± SD	CI (95%)	Mean ± SD	CI (95%)	Mean ± SD	CI (95%)	Mean ± SD	CI (95%)
SW28	1.54 ± 2.32	0.51–2.58	0.91 ± 1.52	0.52–1.31	2.28 ± 3.45	0.29–4.28	1.2 ± 1.86	0.17–2.23
TEN28	1.54 ± 3.02	0.21–2.88	0.98 ± 2.47	0.34–1.63	4.43 ± 8.07	−0.23–9.09	1.71 ± 2.67	0.17–3.26
DAS28	3.3 ± 1.1	2.8–3.8	3.4 ± 1.2	3.08–3.72	3.87 ± 1.51	2.99–4.74	2.54 ± 1.44	1.71–3.37
PGA	34.09 ± 22.37	23.91–44.28	39.60 ± 23.93	33.31–45.89	52.5 ± 21.73	39.95–65.04	42.33 ± 26.45	27.69–56.98
HAQ	0.89 ± 0.86	0.48–1.31	1.09 ± 0.71	0.90–1.28	1.59 ± 0.61	1.23–1.94	1.23 ± 0.76	0.79–1.68
CRP (mg/L)	8.67 ± 8.99	4.34–13.0	6.49 ± 8.8	4.05–8.94	9.4 ± 7.44	5.1–13.7	3.19 ± 3.16	1.44–4.94
Platelets (×10^9^ cells/L)	262.31 ± 107.29	214.74–309.88	262.03 ± 86.53	239.48–284.58	255.5 ± 90.2	203.42–307.58	222.53 ± 39.78	200.5–244.56
% Erithrocytes	40.26 ± 3.51	38.7–41.81	40.2 ± 4.3	39.08–41.32	41.6 ± 4.55	38.97–44.23	40.48 ± 4.36	37.72–43.96
Haemoglobin (g/dL)	13.55 ± 1.07	13.07–14.02	13.42 ± 1.48	13.03–13.80	13.68 ± 1.19	12.99–14.37	14.17 ± 1.62	13.28–15.07
ESR (mm/h)	30.14 ± 13.33	24.23–36.05	35.98 ± 25.35	29.38–42.59	26.64 ± 19.23	15.54–37.75	8.73 ± 8.28	4.15–13.32

## Data Availability

The data presented in this study are available on request from the corresponding author if the content in their original published articles does not cover the information required.

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
