# Peer review of "Soluble CD26: From Suggested Biomarker for Cancer Diagnosis to Plausible Marker for Dynamic Monitoring of Immunotherapy"

_cancers, 2024, doi:10.3390/cancers16132427_

Round 1

Reviewer 1 Report

Comments and Suggestions for Authors

I was invited to read a review by Cordero et al about soluble CD26 as a biomarker for cancer and monitorer for immunotherapy. 

Here are my comments:

- in the simple summary there is nothing said about CD26 though this should be the center of the topic (?)

- chapter 1 and 2 describes CD26 on solid tumours and leukocytes but the focus should be on soluble CD26. So tell here how CD26 is shed by which in solid tumours and how CD26 here function as a biomarker either pro- or anti-tumorgenic/prognostic value

- in the following, authors only describe CD26 as a marker in colorectal cancer. But there are certainly other studies done on solid and non-solid cancers analyzing CD26 as a biomarker

- beside a small paragraph about immunotherapy of rheumatoid arthritis I do not see anything about immunomonitoring of cancer surveillance.

The structure and content of this review is not clear to me. The text does not reflect the title and vice versa. So either change the title or change the entire text, the latter which should focus on biomarker and immunotherapy monitoring in cancer. Non solid cancers such as leukemias are also left out.  Authors and thus also readers got lost in things about CD26 which are not subject to this review – if I correctly understood this work. From time to time, authors bring their own (preliminary) results into this review, it seems like the review format is used to present own results without the need for producing an original study. 

Comments on the Quality of English Language

Grammar and style must be corrected by a naïve. 

Author Response

Please find enclosed a revised version of our manuscript “Soluble CD26: From suggested biomarker for cancer diagnosis to plausible marker for dynamic monitoring of immunotherapy". The revised manuscript has been modified according to the reviewers’ comments and a point-by-point response to their comments is enclosed.

Reply to Reviewers Comments:

Thank you very much to the reviewers for their critics and suggestions, we feel that our manuscript has been greatly improved after their careful consideration.

We will answer some general criticisms first and later the specific of each reviewer.

1) It seems that the review was losing focus along the text, mostly because the molecule is involved in many key functions.

+ In this sense, we reorganized the subheadings and expanded their titles, with the aim of helping to explain the topic of the review: 1. Introduction; 2. Basics of the CD26 protein and soluble CD26 (sCD26); 3. CD26 in solid tumors and why sCD26 was studied as biomarker; 4. CD26 in leukocytes; 5. sCD26 from leukocytes; 6. Studies of sCD26 levels’ changes in patient monitoring; 7. sCD26 shedding and/or secretion: biochemistry of sCD26; 8. CD26 substrates and inhibitors; 9. ADA; 10. sCD26 in saliva; 11. Conclusion.

+ “In the simple summary there was nothing said about CD26 though this should be the center of the topic” (reviewer 4). In our objective of avoiding repetitions in the Simple Summary with respect to the Abstract, we focused on dynamic monitoring in the first and in the biomarker in the second.

We agree with the reviewer that it would be useful to comment on CD26 in the simple summary; therefore, we added in the simple summary the sentence “Immune Checkpoint Immunotherapy alters the local (solid tumor infiltration) and systemic balance among several immune system cell populations. Therefore, we discuss the usefulness of immune cell-related soluble CD26 and its DPP4 activity in immunotherapy monitoring”.

We also added to the Abstract this sentence: “We propose that dynamic monitoring of sCD26/DPP4 changes, in addition to well-known inflammatory biomarkers such as CRP, may indicate resistance or response during the successive steps of the treatment”.

2) Apologies for the issues with the bibliography, due to the specific template of this journal. This has been corrected in the current version of the manuscript.

3) Somehow, it was unclear whether the figures were taken from a previous publication or were data provided for the first time.

We did review not just our work, all data from other authors supporting our proposal was used. Apologies for the misunderstanding, all figures refer to our own data. For clarity, we wanted to illustrate the text with figures and we owned the copyright, as follows. Figure 1 was taken directly from two articles [45,46]. Figures 2 (and now the new Fig 3) and 4 show published data [117, 68] but in a different way, related to the topic of the manuscript. Figure 4 (now 5) was not published before, although the knowledge illustrated in the figure is well known. Having the western blot pictures, and since the point discussed was not at all important for the review, we still think that the discussion on that point is better with the figure.

References citing the source have been added to the figure legends. For Figure 4 (now 5), a description of the method was already made in the legend.

4) Grammar and style have been corrected.

Responses to Reviewer 4's Comments

Point 1. “in the simple summary there is nothing said about CD26 though this should be the center of the topic (?)”.

Answer: Apologies. Please see the general comment.

Point 2. “chapter 1 and 2 describes CD26 on solid tumours and leukocytes but the focus should be on soluble CD26. So tell here how CD26 is shed by which in solid tumours and how CD26 here function as a biomarker either pro- or anti-tumorgenic/prognostic value”

“The structure and content of this review is not clear to me. The text does not reflect the title and vice versa. So either change the title or change the entire text, the latter which should focus on biomarker and immunotherapy monitoring in cancer. Non solid cancers such as leukemias are also left out.  Authors and thus also readers got lost in things about CD26 which are not subject to this review – if I correctly understood this work”.

Answer: The other reviewers more or less agreed with this point of view, that manuscript’s focus was difficult to follow, so all changes explained in the general comment and specific comments from the other reviewers were made to improve the structure and content and so to focus on the biomarker and immunotherapy monitoring in cancer (see point 3 of general reply).

Non solid cancers such as leukemias were left out for this reason. Solid cancers are the focus of the Special Issue and where the story of the biomarker started.

Point 3. “in the following, authors only describe CD26 as a marker in colorectal cancer. But there are certainly other studies done on solid and non-solid cancers analyzing CD26 as a biomarker”.

Point 4. “beside a small paragraph about immunotherapy of rheumatoid arthritis I do not see anything about immunomonitoring of cancer surveillance”. 

Answer to both: The discussion about sCD26/DPP4 begins with its history primarily as diagnostic biomarker, not only for cancer. However, that topic has been extensively reviewed, so we soon moved on to focus on outcomes as monitoring biomarker. Our cancer surveillance results were on colorectal cancer, although we also highlighted the different behavior of sCD26 titers from early to the metastatic stages. This part has been improved by the new Figure 2.  

            We then show our results on immunomonitoring of rheumatoid arthritis. We have added Table 2 to highlight this part. Interesting studies on immunomonitoring of bowel diseases from other authors were also reviewed.

            To our knowledge, there is only one study with sCD26 on immunotherapy monitoring in cancer, and the therapy is not an Immune Checkpoint inhibitor. Moreover, we discuss that this biomarker can be very informative about migratory changes (including tumour infiltration) in immune cell subsets. Current biomarkers CRP and ADA are related in different ways, as at least chemokine CXCL10, to our proposal.

Point 5. “From time to time, authors bring their own (preliminary) results into this review, it seems like the review format is used to present own results without the need for producing an original study”. 

Answer: Apologies if we did not make that point clear, as it happened with other reviewer’s comments. We show only published results, and not preliminary ones, with the only exception is Figure 5, which we added simply because it is related to the topic. We had two goals with this review: First, most current studies try to find prognostic biomarkers that can help classify patients for personalised therapy. However, surveillance studies can be very informative and more practical. Secondly, obviously our next step will be to have retrospective studies that can support sCD26 as immunotherapy biomarker, alone or in a small group of them (multiplexed), but if at least research groups familiarized with the biomarker could conduct a series of studies, this line of research would advance more quickly.

Reviewer 2 Report

Comments and Suggestions for Authors

This is a well-researched article and draws together alot of information in the field of DPP4 biology. The writing will need some work for grammatical errors and English expression, but otherwise this is an informative and entertaining read.

My comments are below.

1.       English expression needs review

2.       Please check the references. References 25-27 are missing from the bibliography, and multiple references in the text do not seem to be correct.

3.       Line 141-143; please specify the type of CAR expressed in cells in this study, otherwise the reference is nonsensical.

4.        Sections 1, 2 and 3 would benefit greatly from the inclusion of descriptive figures to accompany the text. Much of the text (for example describing pathways) could reasonably be substituted into figures instead.

5.       Is data in figures 1 through 4 taken from a previous publication or is it new data provided for the first time? If the former, a reference should be added in the figure legend citing the source. If the latter a description of methods etc is required.

6.       In section 4 “sCD26 shedding and/or secretion: biochemistry of sCD26” proteolytic variants of DPP4 are discussed. There is at least one recent paper (doi: 10.3390/ijms21218110) describing the proteolytic release of DPP4 from cancer cells in an inactivated form, which suggests a reason why activity versus serum titres are often unrelated. This provides an important link between this section and the previous one in section 3 regarding DPP4 as a biomarker of disease, and is worth including.

7.       line 463 – when discussing conversion of chemokines into antagonist forms by DPP4, it should be noted that the evidence for these is primarily in vitro only. There is weak to no evidence for their processing in vivo for most of these.

8.       Section 5 DPP4 substrates and inhibitors – there are several potential impacts of binding noted. However, it should also be included that DPP4i can bind to lymphocytes that are CD26+ with unknown consequence, and that this is also likely to have biological relevance.

9.       Line 469 – delete this sentence, it looks like a mistake left in from something else?

Comments on the Quality of English Language

English expression needs review and work. References also need to be checked and corrected.

Author Response

Please find enclosed a revised version of our manuscript “Soluble CD26: From suggested biomarker for cancer diagnosis to plausible marker for dynamic monitoring of immunotherapy". The revised manuscript has been modified according to the reviewers’ comments and a point-by-point response to their comments is enclosed.

Reply to Reviewers Comments:

Thank you very much to the reviewers for their critics and suggestions, we feel that our manuscript has been greatly improved after their careful consideration.

We will answer some general criticisms first and later the specific of each reviewer.

1) It seems that the review was losing focus along the text, mostly because the molecule is involved in many key functions.

+ In this sense, we reorganized the subheadings and expanded their titles, with the aim of helping to explain the topic of the review: 1. Introduction; 2. Basics of the CD26 protein and soluble CD26 (sCD26); 3. CD26 in solid tumors and why sCD26 was studied as biomarker; 4. CD26 in leukocytes; 5. sCD26 from leukocytes; 6. Studies of sCD26 levels’ changes in patient monitoring; 7. sCD26 shedding and/or secretion: biochemistry of sCD26; 8. CD26 substrates and inhibitors; 9. ADA; 10. sCD26 in saliva; 11. Conclusion.

+ “In the simple summary there was nothing said about CD26 though this should be the center of the topic” (reviewer 4). In our objective of avoiding repetitions in the Simple Summary with respect to the Abstract, we focused on dynamic monitoring in the first and in the biomarker in the second.

We agree with the reviewer that it would be useful to comment on CD26 in the simple summary; therefore, we added in the simple summary the sentence “Immune Checkpoint Immunotherapy alters the local (solid tumor infiltration) and systemic balance among several immune system cell populations. Therefore, we discuss the usefulness of immune cell-related soluble CD26 and its DPP4 activity in immunotherapy monitoring”.

We also added to the Abstract this sentence: “We propose that dynamic monitoring of sCD26/DPP4 changes, in addition to well-known inflammatory biomarkers such as CRP, may indicate resistance or response during the successive steps of the treatment”.

2) Apologies for the issues with the bibliography, due to the specific template of this journal. This has been corrected in the current version of the manuscript.

3) Somehow, it was unclear whether the figures were taken from a previous publication or were data provided for the first time.

We did review not just our work, all data from other authors supporting our proposal was used. Apologies for the misunderstanding, all figures refer to our own data. For clarity, we wanted to illustrate the text with figures and we owned the copyright, as follows. Figure 1 was taken directly from two articles [45,46]. Figures 2 (and now the new Fig 3) and 4 show published data [117, 68] but in a different way, related to the topic of the manuscript. Figure 4 (now 5) was not published before, although the knowledge illustrated in the figure is well known. Having the western blot pictures, and since the point discussed was not at all important for the review, we still think that the discussion on that point is better with the figure.

References citing the source have been added to the figure legends. For Figure 4 (now 5), a description of the method was already made in the legend.

4) Grammar and style have been corrected.

Responses to Reviewer 1's Comments

Points 1-3, 5, 8, 9

  1. English expression needs review
  2. Please check the references. References 25-27 are missing from the bibliography, and multiple references in the text do not seem to be correct.
  3. Line 141-143; please specify the type of CAR expressed in cells in this study, otherwise the reference is nonsensical.
  4. Is data in figures 1 through 4 taken from a previous publication or is it new data provided for the first time? If the former, a reference should be added in the figure legend citing the source. If the latter a description of methods etc is required.
  5. Section 5 DPP4 substrates and inhibitors – there are several potential impacts of binding noted. However, it should also be included that DPP4i can bind to lymphocytes that are CD26+ with unknown consequence, and that this is also likely to have biological relevance.
  6. Line 469 – delete this sentence, it looks like a mistake left in from something else?

Answer to all: Agreed, as it was explained above.

Point 4 “Sections 1, 2 and 3 would benefit greatly from the inclusion of descriptive figures to accompany the text. Much of the text (for example describing pathways) could reasonably be substituted into figures instead.

Answer: We understand the reviewer’s point of view. Since the content of these sections has been subjected to many reviews, and this molecule is involved in so many key pathways, making it difficult to maintain the focus of the manuscript (which should be on soluble CD26, as other reviewers mentioned, see Reviewer 4), we reorganized this part, with the figures intended to support the main topic.

Point 6 “the proteolytic release of DPP4 from cancer cells in an inactivated form, which suggests a reason why activity versus serum titres are often unrelated”.

Answer: We appreciate the reviewer’s suggestion. Although we were aware of the reference, it was left behind. Because it actually “provides an important link between sections” we have included four (with the suggested one) new references [129-132].

Point 7 “when discussing conversion of chemokines into antagonist forms by DPP4, it should be noted that the evidence for these is primarily in vitro only”

Answer: We incorporated the reviewer’s comment to our manuscript. Additionally, we also discuss a case of truncated chemokine in vivo: “For the purpose of the Special Issue, we note here that some of the chemokines truncated by DPP4 can be measured. CXCL10 can be processed to an inactive form in ovarian cancer and the group of Stephens et al. evaluated the ratio of active: total CXCL10 to improve the identification of malignancy, and recently the addition of this test to a biomarker panel including also pro-inflammatory IL6, cancer antigen 125 (CA125) and Human Epididymal Protein 4 (HE4) measurements that are part of the Risk of Malignancy Index (RMI) or Risk of Malignancy Algorithm (ROMA) scores, used to indicate likelihood of malignancy when an ovarian mass is present, improved strongly the discrimination of benign from malignant disease [139,140]. We suggest that multiplexing with this chemokine, truncated or not, and IL6, may enhance the specificity of the test proposed.

Reviewer 3 Report

Comments and Suggestions for Authors

TThis article presents various changes in CD26 and related molecules in cancer and other diseases. The role of CD26 in cancer was previously reported by the authors in the same journal. Indeed, the authors point to the importance of CD26 and related proteins as cancer biomarkers, as shown by their own and other researchers' results. However, it is not clear whether CD26 has any real value as a biomarker for colorectal cancer or other cancers, as the data are not convincing. It is also quite difficult for the reader to understand the authors' true views based on the confirmed data.

Author Response

Please find enclosed a revised version of our manuscript “Soluble CD26: From suggested biomarker for cancer diagnosis to plausible marker for dynamic monitoring of immunotherapy". The revised manuscript has been modified according to the reviewers’ comments and a point-by-point response to their comments is enclosed.

Reply to Reviewers Comments:

Thank you very much to the reviewers for their critics and suggestions, we feel that our manuscript has been greatly improved after their careful consideration.

We will answer some general criticisms first and later the specific of each reviewer.

1) It seems that the review was losing focus along the text, mostly because the molecule is involved in many key functions.

+ In this sense, we reorganized the subheadings and expanded their titles, with the aim of helping to explain the topic of the review: 1. Introduction; 2. Basics of the CD26 protein and soluble CD26 (sCD26); 3. CD26 in solid tumors and why sCD26 was studied as biomarker; 4. CD26 in leukocytes; 5. sCD26 from leukocytes; 6. Studies of sCD26 levels’ changes in patient monitoring; 7. sCD26 shedding and/or secretion: biochemistry of sCD26; 8. CD26 substrates and inhibitors; 9. ADA; 10. sCD26 in saliva; 11. Conclusion.

+ “In the simple summary there was nothing said about CD26 though this should be the center of the topic” (reviewer 4). In our objective of avoiding repetitions in the Simple Summary with respect to the Abstract, we focused on dynamic monitoring in the first and in the biomarker in the second.

We agree with the reviewer that it would be useful to comment on CD26 in the simple summary; therefore, we added in the simple summary the sentence “Immune Checkpoint Immunotherapy alters the local (solid tumor infiltration) and systemic balance among several immune system cell populations. Therefore, we discuss the usefulness of immune cell-related soluble CD26 and its DPP4 activity in immunotherapy monitoring”.

We also added to the Abstract this sentence: “We propose that dynamic monitoring of sCD26/DPP4 changes, in addition to well-known inflammatory biomarkers such as CRP, may indicate resistance or response during the successive steps of the treatment”.

2) Apologies for the issues with the bibliography, due to the specific template of this journal. This has been corrected in the current version of the manuscript.

3) Somehow, it was unclear whether the figures were taken from a previous publication or were data provided for the first time.

We did review not just our work, all data from other authors supporting our proposal was used. Apologies for the misunderstanding, all figures refer to our own data. For clarity, we wanted to illustrate the text with figures and we owned the copyright, as follows. Figure 1 was taken directly from two articles [45,46]. Figures 2 (and now the new Fig 3) and 4 show published data [117, 68] but in a different way, related to the topic of the manuscript. Figure 4 (now 5) was not published before, although the knowledge illustrated in the figure is well known. Having the western blot pictures, and since the point discussed was not at all important for the review, we still think that the discussion on that point is better with the figure.

References citing the source have been added to the figure legends. For Figure 4 (now 5), a description of the method was already made in the legend.

4) Grammar and style have been corrected.

Responses to Reviewer 2's Comments:

Point 1 “It is also quite difficult for the reader to understand the authors' true views based on the confirmed data”.

Answer: We hope that the many changes in the text can help, together with the next answer.

Point 2 “Indeed, the authors point to the importance of CD26 and related proteins as cancer biomarkers, as shown by their own and other researchers' results. However, it is not clear whether CD26 has any real value as a biomarker for colorectal cancer or other cancers, as the data are not convincing.”

Answer: We tried to clearly differentiate between cell surface (plasma membrane) CD26 as biomarker from the soluble CD26 form, which is more related to immune cells. sCD26 has been widely studied primarily as a diagnostic marker. As happens with many biomarkers, specificity needs to be managed but, as our recent article comparing it to FIT shows, it has high sensitivity for the early detection of colorectal cancer. ICI alters the local (solid tumor infiltration) and also systemic balance among several immune system cell populations, and the data included in the review strongly support the relationship between these cell populations and the sCD26 and/or DPP4 activity. We also mention some additional biomarkers that could be multiplexed with sCD26 for dynamic monitoring of therapy (there are many more studies working with prognostic biomarkers, to choose the best personalized therapy, but patients should always be monitored).

Reviewer 4 Report

Comments and Suggestions for Authors

“Soluble CD26: From suggested biomarker for cancer diagnosis to plausible marker for dynamic monitoring of immunotherapy” by Martin Kotrulev, Iria Gomez Touriño and Oscar J. Cordero is a manuscript that proposes an extensive review on a very suggestive molecule, CD26, that can be present on the membrane of several cell types including tumour and immune cells. But can also be found as secreted protein in many body fluids, this soluble form may also show various degrees of glycosylation and be associated to other proteins.  

The manuscript is relatively well written but because this molecule is involved in many key pathways, that go through migration, inflammation, survival and activation the authors touched so many pathologies that the review losses the focus.  Moreover, authors made a mix between their own data and a revision of the literature, leading to a bit of confusion. It seems that figures relate to unpublished results obtain by the authors, but if this is true, at this point details are missing as to figures/figure legends accuracy; and why not to show also results on analysis of CD26 expression on T cell subset etc….(lines 328-335)

Figures and figure legends:

Figure 1: Columns are probably means… authors should add statistics and standard deviation. The number of cases analysed on the first panel are very different, are authors considering to increase the number in the groups less represented. What about increasing also the number of observations in panel B? How levels of sCD26 relate to sex and age in healthy donors? Consider to add this information in figure 1 as starting point.

Figure 2: More than a representative example it would be more convincing to show graphs with the analysis of several patients. Panel C please add the number of CRC patients recruited in the study.

Figure 3: insert on the graph the types of treatment/groups instead of 1,2… In the figure legend the numbers of the groups finish at number 4… Maybe it would be worth it to have information on the disease score, so that we perceive the effects of the inflammatory status of the patients that have been involved in the study.

Missing information: Demographic and clinical characteristics of the individuals recruited in this study.

Comments on the Quality of English Language

Quality of the english can be improved.

Author Response

Please find enclosed a revised version of our manuscript “Soluble CD26: From suggested biomarker for cancer diagnosis to plausible marker for dynamic monitoring of immunotherapy". The revised manuscript has been modified according to the reviewers’ comments and a point-by-point response to their comments is enclosed.

Reply to Reviewers Comments:

Thank you very much to the reviewers for their critics and suggestions, we feel that our manuscript has been greatly improved after their careful consideration.

We will answer some general criticisms first and later the specific of each reviewer.

1) It seems that the review was losing focus along the text, mostly because the molecule is involved in many key functions.

+ In this sense, we reorganized the subheadings and expanded their titles, with the aim of helping to explain the topic of the review: 1. Introduction; 2. Basics of the CD26 protein and soluble CD26 (sCD26); 3. CD26 in solid tumors and why sCD26 was studied as biomarker; 4. CD26 in leukocytes; 5. sCD26 from leukocytes; 6. Studies of sCD26 levels’ changes in patient monitoring; 7. sCD26 shedding and/or secretion: biochemistry of sCD26; 8. CD26 substrates and inhibitors; 9. ADA; 10. sCD26 in saliva; 11. Conclusion.

+ “In the simple summary there was nothing said about CD26 though this should be the center of the topic” (reviewer 4). In our objective of avoiding repetitions in the Simple Summary with respect to the Abstract, we focused on dynamic monitoring in the first and in the biomarker in the second.

We agree with the reviewer that it would be useful to comment on CD26 in the simple summary; therefore, we added in the simple summary the sentence “Immune Checkpoint Immunotherapy alters the local (solid tumor infiltration) and systemic balance among several immune system cell populations. Therefore, we discuss the usefulness of immune cell-related soluble CD26 and its DPP4 activity in immunotherapy monitoring”.

We also added to the Abstract this sentence: “We propose that dynamic monitoring of sCD26/DPP4 changes, in addition to well-known inflammatory biomarkers such as CRP, may indicate resistance or response during the successive steps of the treatment”.

2) Apologies for the issues with the bibliography, due to the specific template of this journal. This has been corrected in the current version of the manuscript.

3) Somehow, it was unclear whether the figures were taken from a previous publication or were data provided for the first time.

We did review not just our work, all data from other authors supporting our proposal was used. Apologies for the misunderstanding, all figures refer to our own data. For clarity, we wanted to illustrate the text with figures and we owned the copyright, as follows. Figure 1 was taken directly from two articles [45,46]. Figures 2 (and now the new Fig 3) and 4 show published data [117, 68] but in a different way, related to the topic of the manuscript. Figure 4 (now 5) was not published before, although the knowledge illustrated in the figure is well known. Having the western blot pictures, and since the point discussed was not at all important for the review, we still think that the discussion on that point is better with the figure.

References citing the source have been added to the figure legends. For Figure 4 (now 5), a description of the method was already made in the legend.

4) Grammar and style have been corrected.

Responses to Reviewer 3's Comments:

Point 1. “the authors touched so many pathologies that the review losses the focus.  Moreover, authors made a mix between their own data and a revision of the literature, leading to a bit of confusion. It seems that figures relate to unpublished results obtain by the authors, but if this is true, at this point details are missing as to figures/figure legends accuracy; and why not to show also results on analysis of CD26 expression on T cell subset etc….(lines 328-335)”

Answer: Thank you for the suggestion. We reply to this issue in the general reply, above, (point 3). Basically, most figures come from published articles with our own data, and this has been stated now also in the Figure legends. Some details were not included to advance quickly through the manuscript text and avoid losing focus, since the original results can always be consulted.

Regarding CD26 on T cells, it would be great for this review if more studies had been conducted on the relationships between the CD26+ and CD26neg immune cell populations and the soluble CD26 protein concentration and DPP4 activity, but this knowledge is still scarce. We feel that Figures with only expression of cell surface CD26 would distract from the main focus.

“Figures and figure legends:

Figure 1: Columns are probably means… authors should add statistics and standard deviation. The number of cases analysed on the first panel are very different, are authors considering to increase the number in the groups less represented. What about increasing also the number of observations in panel B? How levels of sCD26 relate to sex and age in healthy donors? Consider to add this information in figure 1 as starting point.”

Answer: Columns in Fig 1A show Sensitivity values (standard deviation does not apply here). Fig 1B were 4 cases of polypectomy. Then, the interest was to show that sCD26 were normalized after that treatment. This same logic was better achieved in the follow-up study for CRC surveillance, with more time points, as seen in the Figure 2 and 3 (old Fig 2 now sorted).

            Thank you for the second suggestion. In our recent article, important data on sCD26 and DPP4 regarding sex and age was shown in a Table 1 where the total population under study included colorectal cancer cases and advanced adenomas. Here, we modified and simplified that Table, excluding those cases (Table 1).

“Figure 2: More than a representative example it would be more convincing to show graphs with the analysis of several patients. Panel C please add the number of CRC patients recruited in the study.”

Answer: More examples are shown in Fig 2, in agreement with the reviewer. For this reason, panel C is now Figure 3 and its information is clearer. The numbers of CRC patients enrolled in the study were already in the main text, but we couldn’t include them in Fig 3 because we did not have the original figure.

“Figure 3: insert on the graph the types of treatment/groups instead of 1,2… In the figure legend the numbers of the groups finish at number 4… Maybe it would be worth it to have information on the disease score, so that we perceive the effects of the inflammatory status of the patients that have been involved in the study”.

Answer: We apologize, there was a typo in the figure legend of now Fig 4. We also agree with this suggestion, and a Table 2 (already published) with disease activity parameters of patients grouped according to their biological (BT) or conventional (no BT) therapies has been included (to perceive the effects of the inflammatory status of the patients involved in the study). For the “Missing information: Demographic and clinical characteristics of the individuals recruited in this study.”, we refer to the original article (reference 68).

Round 2

Reviewer 1 Report

Comments and Suggestions for Authors

See attachement.

Comments on the Quality of English Language

Please rephrase subheadings, the english language is not sound

Author Response

Manuscript ID: cancers- 2980435

Title:  Soluble CD26: From suggested biomarker for cancer diagnosis to plausible marker for dynamic monitoring of immunotherapy

Journal: Cancers

Please find enclosed the second revised version of our manuscript “Soluble CD26: From suggested biomarker for cancer diagnosis to plausible marker for dynamic monitoring of immunotherapy". The revised manuscript has been modified according to the reviewer 1’ comments, and a point-by-point response to their comments is enclosed.

Reply to Reviewer 1’s Comments:

Thank you very much to the reviewer for their critics and suggestions, our manuscript has been greatly improved after their careful consideration.

  1. Introduction: the focus of the review is soluble CD26, thus the intro should start with the overall role of soluble CD26 followed by its role among biomarkers, and for what

Answer: We agree, but we think that this focus can actually be read in the manuscript. After a very brief introduction explaining why a protein biomarker is informative, the second chapter was titled “Basics of the proteins CD26 and soluble CD26 (sCD26)” to explain the overall role of sCD26, only with a short introduction of CD26 as a refence. Chapter 3 “CD26 in solid tumors and why sCD26 was studied as a cancer diagnostic biomarker” presented sCD26 as a tumor biomarker; at the time it was thought that changes in its blood levels was related to the pro- or anti-tumorigenic roles of CD26 in tumor cells (lower levels -except in metastases- of colorectal cancer CD26 protein correlated with lower levels of sCD26 in blood).

       In parallel with these advances, the acquisition of knowledge on the CD26 presence on the immune system cells accelerated (Chapter 4 “Expression of cell membrane CD26 on leukocytes”). In T lymphocytes, CD26 was originally described as a marker of T activation, although this idea has evolved over time. Based on that idea, there were many studies linking sCD26 levels with a Th1 response in many diseases. At this stage, knowledge on the sCD26 physiology was still scarce, so the idea of Chapter 5 (sCD26 from leukocytes. Relation with specific populations) was to pack there this “immunological” sCD26 knowledge, separated from the previous chapter because it was a different context.

       It should be noted that until the publication of the position paper “The hallmarks of cancer” (Hanahan and Weinberg Cell 100:57–70, 2000), and for several years after, little or no clinical data of immunological origin were recorded, and most of the biomarkers in clinical use were tumor-originated neo-antigens. These are very specific but not helpful for early detection. We showed that sCD26 detects most colorectal advanced adenomas, for example. With this chapter, readers will be now prepared to understand why changes in sCD26 levels may reflect alterations in the balance among immune system populations, although the presence of additional sCD26 from endothelium and other cells such as cancer stem cells cannot be ruled out.

  1. Please rephrase subheadings, the english language is not sound

Answer: Apologies. In addition to the rephrased titles highlighted above, Chapter 6 title “Studies of sCD26 levels’ changes in patients’ monitoring” is not sound (Now, Studies that support changes in sCD26 levels as a tool for patient monitoring). Once the previous topic, biomarker sCD26 from leukocytic origin, was established, we review in this chapter the studies that have used sCD26 for disease monitoring, one for the surveillance of cancer recurrence, another for cancer immunotherapy, and others for immunotherapy of autoimmune diseases. This Chapter is the core of the manuscript. Since it keeps sCD26 in the title, just like in chapter 2, 3 and then 5, we hope readers can follow this line of thinking. At this point in the manuscript the reader can understand why the levels of this biomarker in the blood change. In the next chapters, the content will refer to other practical approaches of the biomarker.

  1. Chapter 7 belongs to the beginning when describing the characteristics of sCD26

Answer: (following the paragraph above) For example, chapter 7. Chapter 7 fit with previous chapters’ content, as reviewer 1 says. However, our idea with this chapter is to expand the information and the possibilities provided by the biomarker. First: “But we want to highlight here the possibility that many different forms of circulating sCD26 exist in blood, each of them related with different types of cells and that eventually could be sorted as more specific biomarkers”.

Second: We suggest that measuring DPP4 activity levels adds information to the sCD26 concentration value. “strong up-regulation of DPP4 mRNA expression under hypoxia growth in cancer cell types including smooth muscle cells, adipocytes and ovarian, and the increased protein levels lead to its proteolytic shedding from the cell surface in an inactivated form mediated by at least MMP10 and MMP13 [129-131]. It has been also reported in hepatoma cells that Serpin B3 induces overexpression of inactive DPP4 [132]. These findings suggest an additional reason why activity versus serum titers is often unrelated.” Without the information in previous chapters, this topic would be more difficult to understand (sentence in Chapter 3: “We also cited many putative explanations to explain the lack of a strong correlation between this DPP4 activity and the sCD26 protein concentration [7,11], and both were used together or separately to show the potential utility of this protein as a marker in the screening, monitoring, and prognosis of some cancers [11,19].”).

       Same with Chapter 8 “CD26 substrates and inhibitors”. On the one hand, regarding the use of DPP4 activity as a biomarker, to remind that “DPP4 has become a relatively novel therapeutic target for many inhibitors (DPP4i), which prolong the half-life of endogenously produced insulin in diabetics [rev 10-12,16,19,72,] by preventinging the degradation of incretins secreted by the enteroendocrine system [136].”. In this case, a discontinuing of this therapy or its substitution should be considered.

       On the other hand, “It is worth noting that the half-life of other growth factors and chemokines is also regulated by sCD26/DPP4 [7,10]”. At least for CXCL10, the DPP4-dependent inactive form and the ratio of active: total CXCL10 have been evaluated to improve the identification of ovarian cancer malignancy (see chapter).

       Consequently, with this Chapter “We suggest that multiplexing with this chemokine, truncated or not, and IL6, may improve the specificity of the test proposed”.

  1. Chapter 4: again, consider meaningful and senseful subheading titles: “CD26 in leukocytes” is not adequate. E.g.: “Expression of CD26 on leukocytes”, unless authors would like to explain the cytoplasmatic function of CD26. Is that so?

Answer: Apologies.  Originally the chapter comprised also what is now Chapter 5, and the title was left. Now, Chapter 4 is titled “Expression of cell membrane CD26 on leukocytes”). The idea of this chapter is not only to understand immune cell surface or intracellular CD26 as the main contributor to blood sCD26 (although the dynamics of its production and plausible roles are not fully understood), but also to recall the numerous ligands described for CD26. This may be an important development for sCD26 as a biomarker in the future, for example ADA (to be mentioned later), or plasminogen bound to sCD26 can be detected (personal communication). These ligands could be marking different origins.

  1. Chapter 5 than describes sCD26 from leukocytes. How much does it overlap to chapter 4?

Answer: Now Chapter 5 “sCD26 from leukocytes. Relation with specific populations”. With the new title we want to highlight, as with Chapter 4, that the different subsets and different production mechanisms of sCD26 could imply future improved specificities, and mainly, that “changes in circulating leukocyte subsets are reflected by changes in sCD26 levels and DPP4 activity.(line 337)

  1. Subheading 6: according to the title of the Ms, it should be here e.g.: “immunomonitoring by sCD26, or, if only cancer is included, e.g.: “Cancer immunomonitoring”

Answer: (Now, “Studies that support changes in sCD26 levels as a tool for patient monitoring”). See point 2 to explanation.

  1. Chapter 8: this chapter should only focus on targeting/inhibiting soluble CD26 and not membrane bound

Answer: The idea of Chapter 8 was explained in point 3. We did not change the title to “soluble CD26 substrates and inhibitors” because, to our knowledge, DPP4i pharmacodynamics with respect to CD26 or sCD26 are not well understood. DPP4i is expected to affect circulating sCD26 more than to cell surface CD26, which is in close proximity to ECM proteins such as collagen and fibronectin, and proteoglycans such as glypicans, with effects at the catalytic site.

  1. Chapter 9: what has ADA to do within this context, also, a few lines are written for chapter 9, is this topic worthwhile mentioning it in this context?

Answer:  Now, ADA, a clinical biomarker related to sCD26.

Apologies. We did not want to oversaturate the reader with mora data, and the fact that ADA activity is a clinical biomarker used for certain diseases (mainly lung-related) was left semi-occult in the manuscript. We rewrote the sentence: “ADA enzymatic activity is a clinical biomarker for certain diseases [143]”. (line 554)

With this in mind, we explain the relationship between sCD26 and ADA, to support the biomarker sCD26 on the one hand, and to point out that the ADA biomarker (alone or in combination) may be especially informative in the context of solid tumors.

Reply to Academic Editor’s Comments:

There is controversy about the value of CD26 as a biomarker, which should be explained clear in the manuscript.

Answer: sCD26 is not a recognized clinical biomarker. However, there are thousands of studies that have analyzed it, or its DPP4 activity, as a biomarker. We discussed extensively in the manuscript that because it is primarily produced by the immune system, it is not a specific diagnostic biomarker, similarly to the CPR protein, or even IL-6, which are used clinically to support diagnostic information to other biomarkers.

We clearly show in the manuscript that “changes in circulating leukocyte subsets are reflected by changes in sCD26 levels and DPP4 activity.” (line 337). The Special Issue topic is “Existing and Emerging Biomarkers for Immune Checkpoint Immunotherapy”. This immunotherapy should alter the balance of those circulating leukocyte populations if there are cells that migrate and infiltrate the tumor.

Reviewer 4 Report

Comments and Suggestions for Authors

Nothing to add. The authors basically addressed all the issues.

Author Response

Thank you very much to the reviewer for their critics and suggestions, our manuscript has been greatly improved after their careful consideration.